# Benchmarking Machine Learning Models for Quantum Error Correction

## Abstract

Quantum Error Correction (QEC) is one of the fundamental problems in quantum computer systems, which aims to detect and correct errors in the data qubits within quantum computers. Due to the presence of unreliable data qubits in existing quantum computers, implementing quantum error correction is a critical step when establishing a stable quantum computer system. Recently, machine learning (ML)-based approaches have been proposed to address this challenge. However, they lack a thorough understanding of quantum error correction. To bridge this research gap, we provide a new perspective to understand machine learning-based QEC in this paper. We find that syndromes in the ancilla qubits result from errors on connected data qubits, and distant ancilla qubits can provide auxiliary information to rule out some incorrect predictions for the data qubits. Therefore, to detect errors in data qubits, we must consider the information present in the long-range ancilla qubits. To the best of our knowledge, machine learning is less explored in the dependency relationship of QEC. To fill the blank, we curate a machine learning benchmark to assess the capacity to capture long-range dependencies for quantum error correction. To provide a comprehensive evaluation, we evaluate seven state-of-the-art deep learning algorithms spanning diverse neural network architectures, such as convolutional neural networks, graph neural networks, and graph transformers. Our exhaustive experiments reveal an enlightening trend: By enlarging the receptive field to exploit information from distant ancilla qubits, the accuracy of QEC significantly improves. For instance, U-Net can improve CNN by a margin of about 50%. Finally, we provide a comprehensive analysis that could inspire future research in this field. The code is available in supplementary material.

## 1 Introduction

Quantum computing (Preskill, 2018; Gyongyosi & Imre, 2019; Wang et al., 2022) is one of the most promising techniques in both computer science and physics. Once fully realized, it can change various fields such as cryptography (Easttom, 2022), material science (Lordi & Nichol, 2021), complex system simulations (Daley et al., 2022), and more. The most significant advantage of quantum computers over traditional computers is their superior computing power, which increases exponentially with the number of qubits (Gill et al., 2022). However, realizing practical and large-scale quantum computers still faces several challenges. Among them, Quantum Error Correction (QEC), aiming to detect and correct errors in data qubits within quantum systems, remains a significant concern. Unlike bit errors in classical computers, quantum errors are more complex (Roffe, 2019; Resch & Karpuzcu, 2021) and arise from varied sources, including environmental noise, imprecise qubit control, and unpredictable qubit interactions. These errors can introduce some inaccuracies in quantum computations, decreasing the overall reliability of the results.

Traditional non-data-driven quantum error correction methods, such as the minimum weight perfect matching (MWPM) (Criger & Ashraf, 2018), encounter scalability challenges on larger quantum systems (Vittal et al., 2023). It is important for error correction schemes to detect errors efficiently within a limited time budget to guarantee quantum computing's practical application. Furthermore, as quantum computers expand in scale, these schemes should work on more quantum bits. Given its computational complexity, MWPM fails to be a choice for QEC in larger quantum computers (Chamberland et al., 2022). On the other hand, recent

advancements have introduced data-driven QEC schemes, which employ neural network architectures like multilayer perceptron (MLP) (Chu et al., 2022) or convolutional neural networks (CNN) (Chamberland et al., 2022) and demonstrate initial success in QEC. However, these machine learning methods are restricted to capture local patterns and cannot model long-range dependency in qubits of quantum systems.

This work focuses on addressing this issue, and the main contribution can be summarized as

- We establish the first comprehensive machine learning benchmark for QEC. Via representing the surface code as a grid or graph structure, we evaluate seven neural architectures for QEC, including various kinds of convolutional neural network, graph neural network, and transformer, etc.

- We are the first to recognize the implicit long-range dependencies within surface code for QEC. This insight paves the way for further research of quantum error correction from this novel perspective.

- We conduct extensive experiments on different scales and error rates to demonstrate the effectiveness of machine learning-based quantum error correction and systematically compare the performance of various approaches. Most neural networks achieve statistically significant improvement over the best existing method. We find the capacity that captures the implicit long-range dependency is significant for QEC.

## 2 Background

### 2.1 Quantum Basics

In classical computers, bits are realized through transistors, which act as electronic switches. A transistor's on/off state represents 1 and 0 to store information, respectively. Differently, in quantum computers, the basic unit of information stored is the quantum bit, also known as the qubit. Unlike bits that are either in the '0' or '1' state in classical computers, a qubit in quantum systems can be written as a quantum superposition of states:

$$|\psi\rangle = \alpha|0\rangle + \beta|1\rangle, \tag{1}$$

where $|0\rangle$ and $|1\rangle$ are the basis states (similar to classical bits 0 and 1), and $\alpha$ and $\beta$ are complex numbers that satisfy the condition $|\alpha|^2 + |\beta|^2 = 1$. A qubit is a superposition of their basic states. To get the computation results, a qubit readout measures the probability of a qubit state $|\psi\rangle$ collapsing into one of its basis states, either $|0\rangle$ (yielding output $y = +1$) or $|1\rangle$ (yielding output $y = -1$). The probabilities of yielding $|0\rangle$ and $|1\rangle$ are $|\alpha|^2$ and $|\beta|^2$, respectively.

For an $n$-qubit system, a state is described by a vector space and can be written as a tensor product:

$$|\psi\rangle = \alpha_0|\underbrace{00\cdots00}_{n}\rangle + \alpha_1|\underbrace{00\cdots01}_{n}\rangle + \cdots + \alpha_{2^n-1}|\underbrace{11\cdots11}_{n}\rangle, \tag{2}$$

where $\sum_{i=0}^{2^n-1} |\alpha_i|^2 = 1$. Thus, a state in the $n$-qubit system is the superposition of the $2^n$ basis states. A complex state vector of length $2^n$, with all combination coefficients, can describe the entire quantum state.

Like classical logic gates in classical computing, computation in a quantum system is realized by quantum gates. Quantum gates are unitary operators performing unitary transformations on the state vector. We can denote quantum gates as unitary matrices. To perform computation in quantum systems, $k$ quantum gates are applied to perform unitary transformation:

$$|\psi(\boldsymbol{x}, \boldsymbol{\theta})\rangle = U_k\left(\boldsymbol{x}, \theta_k\right) \cdots U_2\left(\boldsymbol{x}, \theta_2\right) U_1\left(\boldsymbol{x}, \theta_1\right)|0\rangle, \tag{3}$$

where $\boldsymbol{x}$ is the input data, $\theta = (\theta_1, \theta_2, \dots)$ are parameters in quantum gates, and $U$ is the unitary matrix. We can embed both the input data and parameters into the quantum state $|\psi(\boldsymbol{x}, \boldsymbol{\theta})\rangle$.

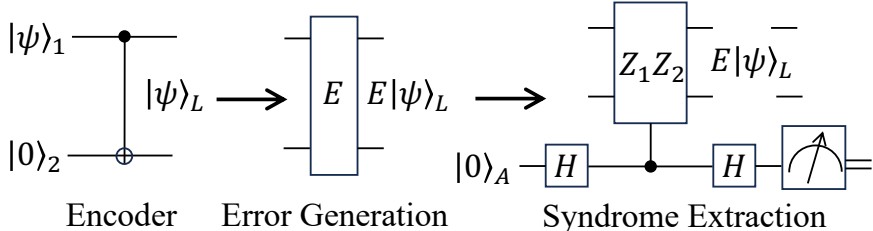

Figure 1: Illustration of syndrome extraction inspired by Roffe (2019). We first add a redundancy qubit $|0\rangle_2$ on $|\psi\rangle_1$ to create a logical state $|\psi\rangle_L$. Then some errors (denoted $E$) are applied on $|\psi\rangle_L$ to get $E|\psi\rangle_L$. After that, we use stabilizer $Z_1Z_2$ controlled by ancilla qubit $A$ to extract the syndrome.

## 2.2 Quantum Errors

Quantum errors can be represented by the Pauli matrices $\boldsymbol{X} = \begin{bmatrix} 0 & 1 \\ 1 & 0 \end{bmatrix}$ and $\boldsymbol{Z} = \begin{bmatrix} 1 & 0 \\ 0 & -1 \end{bmatrix}$ (Roffe, 2019). In general, we have two error-types: $X$-error and $Z$-error, which can be written as follows:

$$\boldsymbol{X}|\psi\rangle = \alpha\boldsymbol{X}|0\rangle + \beta\boldsymbol{X}|1\rangle = \alpha|1\rangle + \beta|0\rangle, \qquad \boldsymbol{Z}|\psi\rangle = \alpha\boldsymbol{Z}|0\rangle + \beta\boldsymbol{Z}|1\rangle = \alpha|0\rangle - \beta|1\rangle. \tag{4}$$

$X$-type errors can be regarded as quantum bit-flips, and $Z$-errors are considered phase-flips. Therefore, the $X$ and $Z$ errors on an $n$-qubit system can be generalized as:

$$\boldsymbol{X}_i|\psi\rangle \text{ and } \boldsymbol{Z}_j|\psi\rangle, \tag{5}$$

where $\boldsymbol{X}_i$ represents an $X$-type error (bit-flip) on the $i$-th qubit and $\boldsymbol{Z}_j$ denotes a $Z$-type error (phase-flip) on the $j$-th qubit. In traditional computer systems, classical bits can be duplicated for error correction. Due to the no-cloning theorem for quantum states (Park, 1970; Wootters & Zurek, 1982), we cannot replicate quantum states for error correction. Besides, in classical systems, we can measure arbitrary properties of the bit register without corrupting the stored information. However, measurements on the qubits during error correction must be meticulously selected for quantum codes since an improper measurement can cause the wavefunction to erase the encoded data (Roffe, 2019).

## 2.3 Quantum Error Correction

Error correction in traditional bits is realized by adding redundancy. Similarly, we can add redundancy in quantum codes to expand the Hilbert space for quantum error correction. Considering $|\psi\rangle = \alpha|0\rangle + \beta|1\rangle$, we use two-qubit encoder to add redundancy and $|\psi\rangle$ can be rewritten as:

$$|\psi\rangle_L = \alpha|00\rangle + \beta|11\rangle = \alpha|0\rangle_L + \beta|1\rangle_L. \tag{6}$$

Therefore, after encoding, a logical qubit can be parameterized as a four-dimensional Hilbert space:

$$|\psi\rangle_L \in \mathcal{H}_4 = \text{span}\{|00\rangle, |01\rangle, |10\rangle, |11\rangle\}. \tag{7}$$

But the logical qubit is defined in a two-dimensional subspace of this extended Hilbert space:

$$|\psi\rangle_L \in \mathcal{C} = \text{span}\{|00\rangle, |11\rangle\} \subset \mathcal{H}_4, \tag{8}$$

where $\mathcal{C}$ is the code space. If we add an $X$-error on the first qubit of $|\psi\rangle_L$, the resulting state is

$$X_1|\psi\rangle_L = \alpha|10\rangle + \beta|01\rangle. \tag{9}$$

Such a state occupies a new subspace:

$$X_1|\psi\rangle_L \in \mathcal{F} \subset \mathcal{H}_4, \tag{10}$$

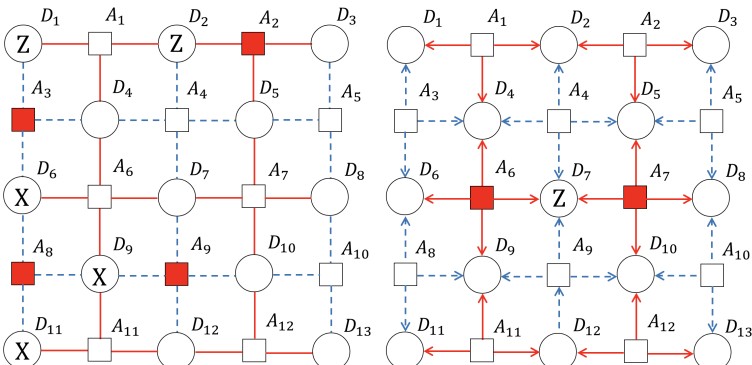

Figure 2: Illustration of surface code (Section 2.4). The circles and squares represent the data and ancilla qubits, respectively. The red solid edges and blue dashed edges represent the controlled-X and controlled-Z operations controlled by ancilla qubits and acting on data qubits, respectively. As shown on the right, only $A_6$ and $A_7$ exhibit syndrome. For instance, $A_1$ does not have the syndrome, meaning the data qubits connected to $A_1$ are error-free. Based on the process of elimination, we can ascertain that only $D_7$ has an error.

where $\mathcal{F}$ is the error subspace. Thus, if the qubit $|\psi\rangle_L$ is not corrupted, it occupies the subspace $\mathcal{C}$. After the $X_1$ error, it occupies the subspace $\mathcal{F}$. Obviously, $\mathcal{C}$ and $\mathcal{F}$ are mutually orthogonal. Consequently, we can use a projective measurement to distinguish which subspace the logical qubit occupies without destroying the encoded quantum information. We can perform a projective measurement $Z_1 Z_2$ on a logical state and get a $(+1)$ eigenvalue:

$$Z_1 Z_2 |\psi\rangle_L = Z_1 Z_2 (\alpha|00\rangle + \beta|11\rangle) = (+1)|\psi\rangle_L, \tag{11}$$

where $Z_1 Z_2$ operator is called stabilizer since it does not change the logical qubit (Gottesman, 2010). But $Z_1 Z_2$ operator projects the errored states, $X_1|\psi\rangle_L$ and $X_2|\psi\rangle_L$ onto the (-1) eigenspace. Note that the coefficients $\alpha$ and $\beta$ are not changed. Then, we use ancilla qubit to extract the syndrome for quantum error correction. The process of quantum error correction is illustrated in Figure 1. First, we add a redundancy qubit to obtain a logical state. After this, some errors are applied to this state. We then utilize stabilizers to extract the syndrome for error correction.

## 2.4 Surface Code

The widely used quantum error correction protocol for experiments is surface code (Fowler et al., 2012; Tomita & Svore, 2014; Roffe, 2019). The surface code is defined on a two-dimensional square lattice (Chamberland et al., 2022) with $(2d-1) \times (2d-1)$ vertices, where $d$ is the distance of the code. Data and ancilla qubits are placed at the vertices, which are connected by the edges. We use an example in Figure 2 to illustrate the structure and elements of a surface code.

In this figure, ancilla qubit $A_2$ is connected to data qubits $D_2, D_3$, and $D_5$ via red edges. These connections signify the underlying relationships between these qubits and allow the ancilla qubit to measure stabilizers such as $X_{D_2}$, $X_{D_3}$, and $X_{D_5}$. When a specific error, for instance, a $Z_{D_2}$ error, occurs on data qubit $D_2$, it anti-commutes with the stabilizer measured by ancilla qubit $A_2$. These $Z$-type errors on the connected data qubits lead to a syndrome in the ancilla qubit $A_2$, which can be identified through the structure of red edges.

Note that a syndrome in an ancilla qubit is caused only by an odd number of the same-type errors acting on that ancilla qubit. This can be demonstrated through another scenario in Figure 2: even though $Z$-type errors occur in data qubits $D_1$ and $D_2$, there is no syndrome in ancilla qubit $A_1$. This absence occurs since the number of errors is even, and their effects cancel out. Besides, the surface code is known to be a degenerate code (Varsamopoulos et al., 2020). This property implies that various sets of errors might generate identical syndromes within the surface code.

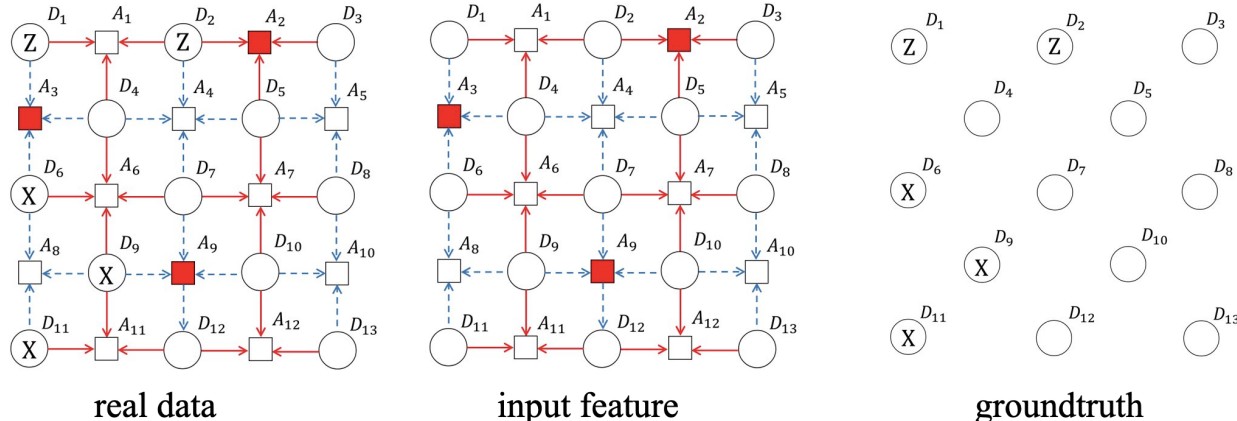

real data             input feature            groundtruth

Figure 3: Formulation of Quantum error correction. (a). Real data contains two kinds of qubits: data qubits (circle) and ancilla qubits (square). (b). Only the ancilla qubits are observed. We use the grid structure and the values of ancilla qubits as the input feature; (c). We are interested in inferring the error types of data qubits (no error, X error, Z error or X&Z error).

## 2.5 Notation

We denote a surface code with distance $d$ as an undirected graph, where $\mathcal{G} = (\mathcal{V}, \mathcal{E})$. $\mathcal{V}$ is the set of nodes $\{v_1, \cdots, v_{(2d-1)^2}\}$ with $|\mathcal{V}| = (2d-1)^2$ and $\mathcal{E} = \{e_{ij}|(i,j) \subseteq |\mathcal{V}| \times |\mathcal{V}|\}$ is the set of edges representing the controlled-$X/Z$ operations, controlled by ancilla qubits and acting on data qubits. The adjacency matrix is defined as $\mathcal{A} \in \{0,1\}^{(2d-1)\times(2d-1)}$, and $\mathcal{A}_{i,j} = 1$ if and only if $(v_i, v_j) \in \mathcal{E}$. Let $\mathcal{N}_k = \{v_j|\mathcal{A}_{k,j} = 1\}$ denote the neighborhood of node $v_k$. We illustrate the surface code in Figure 3. Given Figure 3, we observe that the number of neighbors $|\mathcal{N}_k|$ for a node $k$ lies in the set $\{2,3,4\}$. Based on certain conditions, the neighbors for the node $k$ can be described as:

$$
\begin{cases}
v_{k-1}, & \text{if } k-1 \geq \lfloor(k-1)/(2d-1)\rfloor(2d-1)+1, \\
v_{k+1}, & \text{if } k+1 \leq \lfloor(k-1)/(2d-1)\rfloor(2d-1)+2d-2, \\
v_{k-(2d-1)}, & \text{if } k-(2d-1) \geq 1, \\
v_{k+(2d-1)}, & \text{if } k+(2d-1) \leq (2d-2)(2d-1).
\end{cases}
\tag{12}
$$

Our objective is to detect and correct errors using the syndrome information. While errors manifest in the data qubits, syndromes are present in the ancilla qubits. As shown in Figure 3, the surface code comprises two distinct node types. With this understanding, we can formulate the problem in the following manner. Considering the syndrome information as features of the ancilla qubits, we predict the presence of $X/Z$ errors in the data qubits. Based on the specific syndrome type, we can define the features:

$$
\mathbf{X}_k = 
\begin{cases}
[0,0,0]^{\mathrm{T}}, & \text{if } k\%2 = 1, \\
[0,0,0]^{\mathrm{T}}, & \text{if } k\%2 = 0 \text{ and no syndrome in } v_k, \\
[0,1,0]^{\mathrm{T}}, & \text{if } \lfloor(k-1)/(2d-1)\rfloor\%2 = 0 \text{ and } k\%2 = 0 \text{ and a syndrome in } v_k, \\
[0,0,1]^{\mathrm{T}}, & \text{if } \lfloor(k-1)/(2d-1)\rfloor\%2 = 1 \text{ and } k\%2 = 0 \text{ and a syndrome in } v_k.
\end{cases}
\tag{13}
$$

where % denotes the modulo operator. The vector $\mathbf{X}_k$ can be described based on various conditions related to data and ancilla qubits, as well as the type of syndrome present. For data qubits (when $k\%2 = 1$), $\mathbf{X}_k = [0,0,0]^{\mathrm{T}}$. For ancilla qubits (when $k\%2 = 0$) with no syndrome in $v_k$, $\mathbf{X}_k = [0,0,0]^{\mathrm{T}}$. We define the feature vector of the first type of syndrome in ancilla qubits (when $k\%2 = 0$), which is the result of a $Z$ error (when $\lfloor(k-1)/(2d-1)\rfloor\%2 = 0$) as $\mathbf{X}_k = [0,1,0]^{\mathrm{T}}$. We define the feature vector of the second type of syndrome, which arises from an $X$ error (when $\lfloor(k-1)/(2d-1)\rfloor\%2 = 1$) in ancilla qubits (when $k\%2 = 0$), as $\mathbf{X}_k = [0,0,1]^{\mathrm{T}}$.

To represent the labels of data qubits (specifically when $k\%2 = 0$), we can categorize based on the errors present in $v_k$ as follows:

$$
\mathbf{Y}_{k|k\%2=0} = \begin{cases} [1,0,0,0]^{\mathrm{T}}, & \text{if there is no error in } v_k, \\ [0,1,0,0]^{\mathrm{T}}, & \text{if X error exists in } v_k, \\ [0,0,1,0]^{\mathrm{T}}, & \text{if Z error exists in } v_k, \\ [0,0,0,1]^{\mathrm{T}}, & \text{if both X and Z errors exist in } v_k. \end{cases} \tag{14}
$$

For data qubits ($k\%2 = 1$), if there is no error in $v_k$, the label vector is defined as $\mathbf{Y}_k = [1,0,0,0]^{\mathrm{T}}$. If an $X$ error is present in $v_k$, the label vector is $\mathbf{Y}_k = [0,1,0,0]^{\mathrm{T}}$. If a $Z$ error is detected in $v_k$, the label vector becomes $\mathbf{Y}_k = [0,0,1,0]^{\mathrm{T}}$. If both $X$ and $Z$ errors exist in $v_k$, the label vector is set as $\mathbf{Y}_k = [0,0,0,1]^{\mathrm{T}}$.

## 2.6  Related Work

While traditional error-correction methods, often referred to as decoders, such as the minimum weight perfect matching (MWPM) (Criger & Ashraf, 2018), have shown promise, they face scalability challenges on larger quantum systems (Vittal et al., 2023). It is important for error correction schemes to detect errors efficiently within a limited time budget to guarantee quantum computing's practical application. Furthermore, as quantum computers expand in scale, these schemes should work on more quantum bits. Given its computational complexity, MWPM fails in larger quantum computers (Chamberland et al., 2022).

To address the limitations of MWPM, recent advancements have introduced data-driven QEC schemes. Most data-driven QEC schemes employ neural network architectures like multilayer perceptron (MLP) (Chu et al., 2022) or convolutional neural networks (CNN) (Chamberland et al., 2022), which have shown success in quantum error correction. In general, these models fall into two categories: low-level and high-level ML-based decoding schemes. Low-level decoders are designed to detect and correct errors in data qubits, whereas high-level decoders aim to correct logical errors incorporated by low-level decoders (Bhoumik et al., 2021; Varsamopoulos et al., 2020). Besides, some scalable and fast decoders (Varsamopoulos et al., 2020; Gicev et al., 2021) have been proposed for large-scale surface codes. Despite these promising advances, current methods exhibit significant limitations, primarily due to a lack of modelling of long-range dependency.

To address this issue, this work introduces a new perspective to understanding QEC to fill this research gap. While syndromes in ancilla qubits traditionally stem from errors in adjacent data qubits within the surface code, our findings indicate that long-range ancilla qubits, which are not immediately adjacent but are farther apart, can offer supplementary information. This can help dismiss certain incorrect predictions for error detection in data qubits. Thus, we rely on the implicit long-range relationship between data qubits and remote ancilla qubits to improve error detection performance. Correspondingly, we curate a benchmark to evaluate the performance of existing machine learning methods to capture these long-range dependencies. More specifically, seven popular deep learning algorithms, including CNN (LeCun et al., 1998), Graph Convolutional Network (GCN) (Kipf & Welling, 2017), Multi-Scale Graph Neural Networks (MultiGNN) (Abu-El-Haija et al., 2020), GCN with Initial residual & Identity mapping (GCNII) (Chen et al., 2020), U-Net (Ronneberger et al., 2015), GraphTransformer (Ying et al., 2021), and Approximate Personalized Propagation of Neural Predictions (APPNP) (Gasteiger et al., 2018), are evaluated in the experiments.

# 3  Implicit Long-range Dependency Learning for Quantum Error Correction

## 3.1  Intuition

In this section, we delve into the underlying intuition of exploiting context information from distant ancilla qubits for more accurate error detection within quantum computing systems. Consider the illustrative example on the right of Figure 2. Within this figure, syndromes are detected in both ancilla qubits $A_6$ and $A_7$. An error occurring within data qubits $D_4, D_6, D_7,$ or $D_9$ could lead to a syndrome in $A_6$. Hence, the syndrome information in $A_6$ is insufficient to deduce an error's position. However, the absence of a syndrome in $A_1$ or $A_{11}$ presents additional critical context information that must not be overlooked. By aggregating this contextual information, that is, recognizing that there is no syndrome in $A_1$ or $A_{11}$, we gain

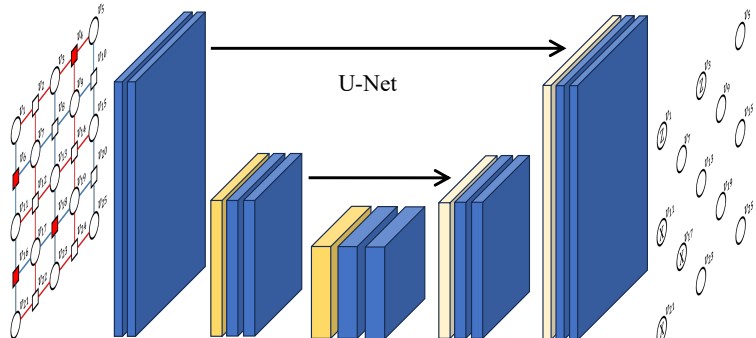

Figure 4: Illustration of U-Net (Ronneberger et al., 2015). By utilizing convolution and downsampling in U-Net, we effectively capture information from distant ancilla qubits for QEC. Notably, the input and output dimensions remain consistent due to upsampling in U-Net.

implicit long-range dependency for error correction in $D_7$. Specifically, this long-range dependency contextual information allows us to confidently rule out the scenario where $D_4$, $D_6$, or $D_9$ are in an error state.

This example serves to highlight the complexity of error detection in quantum systems. It is not merely about identifying where syndromes have occurred but also about understanding where they have not. This recognition can be helpful for more precise error detection, inspiring future research for more effective error correction strategies.

### 3.2 Probabilistic View

In this section, we explore a probabilistic perspective to further our understanding of quantum error correction. Drawing inspiration from machine learning models, we formulate a mathematical framework that integrates both nearby and distant syndrome information, thereby enhancing prediction capacity. For ease of exposition, we list the mathematical notations in Table 4 in Appendix.

Machine learning (ML) models for quantum error correction can typically be construed as learned classification functions. The presence of a syndrome in ancilla qubits is directly determined by the error conditions in the adjacent data qubits. Thus, the main objective of these models is to predict by estimating the posterior distribution $P_\Theta \left( \mathbf{Y}_k \mid \mathbf{X}_{\mathcal{N}_k} \right)$ based on the proximate syndrome data, denoted as $\mathbf{X}_{\mathcal{N}_k}$. In general, these models utilize Maximum Likelihood Estimation (MLE) (Casella & Berger, 2021) to deduce the learnable parameter $\Theta$ (e.g., weight matrix in neural network) by refining the ensuing likelihood function:

$$\arg\max_{\Theta} \prod_k P_\Theta \left( \mathbf{Y}_k | \mathbf{X}_{\mathcal{N}_k} \right), \tag{15}$$

where $k$ symbolizes the $k$-th data qubit in the training dataset, and $\mathbf{X}_{\mathcal{N}_k}$ denotes the syndrome information in the nearby ancilla qubits of the $k$-th data qubit. Building on our previous discussions, it becomes evident that tapping into implicit long-range dependencies can enhance error detection accuracy in data qubits. To exploit this potential, we formulate a context-aware model for quantum error correction designed to leverage these long-range dependency cues. This model integrates the syndrome information, $\mathbf{X}_{v_i|d_{ik}>1}$, from distant ancilla qubits into the likelihood formulation. Consequently, the MLE is restructured as:

$$\arg\max_{\Theta} \prod_k P_\Theta \left( \mathbf{Y}_k | \mathbf{X}_{\mathcal{N}_k}, \mathbf{X}_{v_i|d_{ik}>1} \right), \tag{16}$$

where $v_i|d_{ik}>1$ denotes the distant neighbors. The key idea of addressing Quantum Error Correction (QEC) lies in determining the extent of syndrome information from distant ancilla qubits required for more refined correction. It is formulated as a supervised learning problem. The learning objective is to minimize the cross entropy loss between predictions and groundtruth for all the data qubits, $\mathcal{L} = \sum_k \sum_j (\mathbf{Y}_k)_j \log \left( \widetilde{\mathbf{Y}}_k \right)_j$, where

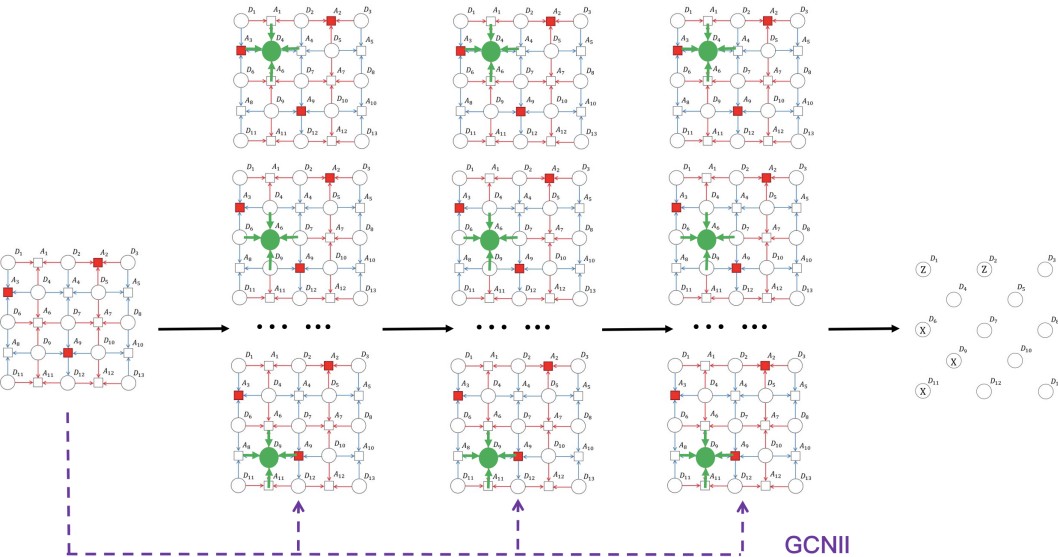

Figure 5: Illustration of GCN (Kipf & Welling, 2016) and GCNII (Chen et al., 2020) (purple dashed line). GCN stacks graph convolution layers and enlarges the receptive field to capture information from distant ancilla qubits for QEC. In each layer of GCN, each node aggregates messages from its neighbors. Based on GCN, GCNII incorporates the raw feature (purple dashed line).

$\widetilde{\mathbf{Y}}_k$ denotes the prediction for the $k$-th data qubit, $\mathbf{Y}_k$ is the groundtruth, $\left(\widetilde{\mathbf{Y}}_k\right)_j$ is the $j$-th element of vector $\widetilde{\mathbf{Y}}_k$. Additionally, the challenge is to employ this information from distant ancilla qubits for QEC efficiently. In the subsequent sections, we will evaluate the efficacy of various context-aware machine learning models designed for QEC.

### 3.3 Methods

As demonstrated by our intuition into the quantum error correction process, the surface code's complexity can be effectively represented as either a 2D grid or a graph. Within this representation, syndromes are manifested by the errors of neighboring data qubits. Besides, by utilizing syndrome information in distant ancilla qubits, we can capture implicit long-range dependency for quantum error correction. Thus, by observing the syndrome and aggregating information from the nearby and distant ancilla qubits, it becomes possible to predict not only the presence of an error within the data qubits but also its specific type. In the following, we introduce seven popular deep learning architectures for quantum error correction, including CNN, U-Net, GCN, GCNII, APPNP, Multi-GNNs, and Graph Transformer.

**Convolutional Neural Network (CNN).** Convolutional neural network (CNN) (O'Shea & Nash, 2015) is a dynamic neural network architecture designed to identify local patterns in input features through the use of convolutional layers, also known as filters or kernels, that slide along the input. It has been used for quantum error correction by Gicev et al. (2021). To keep the size of the surface code unchanged during convolution, we employ a $3 \times 3$ kernel.

**U-Net.** U-Net (Ronneberger et al., 2015), illustrated in Figure 4, is an extension of convolutional neural network architecture used primarily for image segmentation tasks in computer vision. It consists of a contracting path to capture context and a symmetric expanding path to enable precise localization. The U-Net leverages encoder-decoder architectures, where the encoder and decoder perform downsampling and upsampling, respectively. It is widely used in medical image analysis and other image segmentation applications (Li et al., 2018). Due to its downsampling and upsampling operations, we can not only aggregate information from distant ancilla qubits but also maintain the consistent size of the surface code.

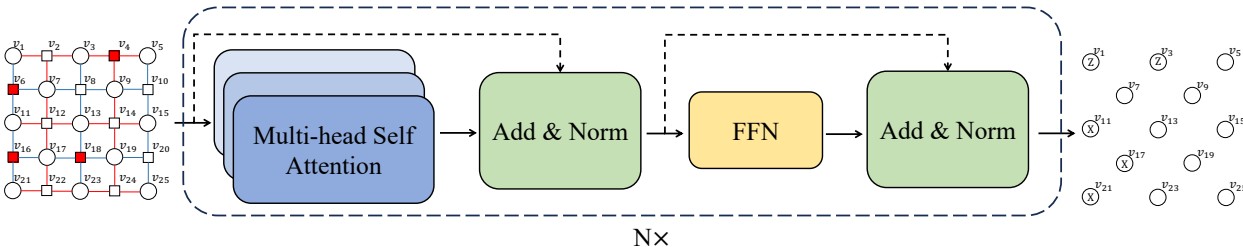

Figure 6: Illustration of Graph Transformer (Ying et al., 2021). By exploiting the self-attention module, we can aggregate information from distant ancilla qubits.

**Graph Convolutional Network (GCN).** Graph convolutional network (GCN) (Kipf & Welling, 2016) is one of the earliest and most widely used graph neural networks. The main idea is to iteratively update the node representation by aggregating the information from its neighbors. The updating rule of GCN for the $l$-th layer is

$$\mathbf{H}^{(l)} = \sigma\big(\widetilde{\mathbf{D}}^{-\frac{1}{2}}\widetilde{\mathbf{\mathcal{A}}}\widetilde{\mathbf{D}}^{-\frac{1}{2}}\mathbf{H}^{(l-1)}\mathbf{W}^{(l)}\big), \tag{17}$$

where $\mathbf{\mathcal{A}} \in \{0,1\}^{|V|\times|V|}$ is the adjacency matrix of the graph $\mathcal{G}$, $\widetilde{\mathbf{\mathcal{A}}} = \mathbf{\mathcal{A}} + \mathbf{I}$, $\mathbf{I}$ is identity matrix, $\widetilde{\mathbf{D}}$ is the diagonal node degree matrix of $\widetilde{\mathbf{\mathcal{A}}}$, $\sigma$ is activation function, e.g., Tanh, ReLU (Rasamoelina et al., 2020). Within the $l$-th layer, $\mathbf{W}^{(l)} \in \mathbb{R}^{d\times d}$ is the learnable weight matrix to transform the embedding, $d$ is the dimension of embedding, $L$ is the depth of GCN. The $i$-th row of $\mathbf{H}^{(l)}$ denotes an embedding vector of the $i$-th node. The illustration of GCN can be found in Figure 5. All the graph neural networks in this paper use the same mathematical notations.

**GCNII.** It is well known that stacking multiple graph convolutional network layers causes over-smoothing, which suggests node representations converge to a similar vector after undergoing multi-layer aggregation. To alleviate this, GCN with Initial residual & Identity mapping (GCNII) (Chen et al., 2020) extends the traditional GCN with initial residual and Identity mapping techniques, as illustrated in Figure 5. The $l$-th layer of GCNII is

$$\mathbf{H}^{(l)} = \sigma\left(\left((1-\alpha_l)\widetilde{\mathbf{D}}^{-\frac{1}{2}}\widetilde{\mathbf{\mathcal{A}}}\widetilde{\mathbf{D}}^{-\frac{1}{2}}\mathbf{H}^{(l-1)} + \alpha_l\mathbf{H}^{(0)}\right)\left((1-\beta_l)\mathbf{I} + \beta_l\mathbf{W}^{(l)}\right)\right), \tag{18}$$

where $\alpha_l$ decides the extent of initial residual connection and $\beta_l$ is for residual connection. The definition of $\sigma, \widetilde{\mathbf{D}}, \mathbf{H}^{(l)}, \widetilde{\mathbf{\mathcal{A}}}, \mathbf{W}^{(l)}$ are same as GCN.

**APPNP.** Approximate personalized propagation of neural predictions (APPNP) (Gasteiger et al., 2018; 2019) is a variant of graph neural network that leverages PageRank-inspired message passing strategy (Rogers, 2002). The main idea of APPNP is to exploit graph diffusion (Gasteiger et al., 2019) to aggregate information from higher-order neighbors. The update rule of the $l$-th layer is

$$\mathbf{H}^{(l)} = \sigma\big((1-\alpha)\widetilde{\mathbf{D}}^{-\frac{1}{2}}\widetilde{\mathbf{\mathcal{A}}}\widetilde{\mathbf{D}}^{-\frac{1}{2}}\mathbf{H}^{(l-1)}\mathbf{W}^{(l)} + \alpha\mathbf{H}^{(0)}\mathbf{W}^{(l)}\big), \tag{19}$$

where $\alpha$ represents transport probability. The definition of $\sigma, \widetilde{\mathbf{D}}, \mathbf{H}^{(l)}, \widetilde{\mathbf{\mathcal{A}}}, \mathbf{W}^{(l)}$ are same as GCN.

**Multi-Scale Graph Neural Networks (Multi-GNNs).** In contrast to APPNP, multi-scale GNNs (Abu-El-Haija et al., 2020) adopt a concatenation-style design to exploit higher-order information from distant ancilla qubits. The architecture can be formulated as:

$$\mathbf{H}^{(l)} = \sigma\big(\big[\mathbf{\mathcal{A}}\mathbf{X}\,|\,\mathbf{\mathcal{A}}^2\mathbf{X}\,|\,\mathbf{\mathcal{A}}^3\mathbf{X}\,|\ldots|\,\mathbf{\mathcal{A}}^l\mathbf{X}\big]\mathbf{W}\big), \tag{20}$$

where $\mathbf{W} \in \mathbb{R}^{d\times c}$ is the trainable weight matrix, $|$ denotes the concatenation. The definition of $\mathbf{H}^{(l)}, \mathbf{\mathcal{A}}, \sigma$ are same as GCN.

Table 1: Overall Accuracy (%). $p$ is the probability of X and Z error in data qubits. OOM is the out-of-memory error. For each task, we highlight the best method in **bold**. We underline the statistically significantly better (pass the t-test, i.e., p-value<0.05) methods than CNN, the best existing method (Chamberland et al., 2022).

| Distance | 3 | | | 5 | | | 7 | | |
|---|---|---|---|---|---|---|---|---|---|
| p | 0.005 | 0.01 | 0.05 | 0.005 | 0.01 | 0.05 | 0.005 | 0.01 | 0.05 |
| MWPM | 97.20 | 96.03 | 88.33 | 85.43 | 83.02 | 76.54 | 84.40 | 81.72 | 78.38 |
| CNN | 99.14 | 98.48 | 91.63 | 99.19 | 98.35 | 91.24 | 99.39 | 98.73 | 91.14 |
| U-Net | **99.94** | **99.78** | **95.23** | 99.95 | 99.81 | **95.88** | **99.96** | **99.83** | **95.98** |
| GCN | 99.72 | 99.66 | 95.01 | 99.94 | 99.77 | 95.72 | 99.94 | 99.79 | 95.69 |
| GCNII | **99.94** | **99.78** | 95.06 | **99.96** | 99.80 | 95.65 | 99.95 | 99.82 | 95.68 |
| APPNP | 99.51 | 98.84 | 90.41 | 99.71 | 99.33 | 91.32 | 99.71 | 99.45 | 92.33 |
| Multi-GNN | 99.80 | 99.52 | 94.39 | 99.87 | 99.51 | 95.08 | 99.93 | 99.70 | 95.07 |
| GraphTransformer | **99.94** | **99.78** | 95.05 | 99.95 | **99.82** | 95.84 | **99.96** | 99.81 | OOM |

**Graph Transformer.** Transformer (Vaswani et al., 2017), illustrated in Figure 6, is one of the most powerful neural structures, finding applications across diverse fields like natural language processing and computer vision. Notably, its prowess extends to graph-structured data, with recent implementations yielding impressive results (Ying et al., 2021). In our study, we delve into the efficacy of the Graph Transformer (Ying et al., 2021) in the context of QEC. A typical Graph Transformer layer comprises a self-attention mechanism coupled with a position-wise feed-forward network (FFN). When given hidden states as input, represented by $\mathbf{H} = \left\{ \mathbf{H}_0^{(l)}, \mathbf{H}_1^{(l)}, \cdots, \mathbf{H}_{(2d-1)^2}^{(l)} \right\}$ for the $l$-th layer of the Graph Transformer, we employ learnable query, key, and value parameter $(\mathbf{W}_Q, \mathbf{W}_K, \mathbf{W}_V)$ to transform $\mathbf{H}^{(l-1)}$ and yield query, key and value matrices:

$$\mathbf{Q}_{(l)} = \mathbf{H}^{(l-1)}\mathbf{W}_Q, \quad \mathbf{K}_{(l)} = \mathbf{H}^{(l-1)}\mathbf{W}_K, \quad \mathbf{V}_{(l)} = \mathbf{H}^{(l-1)}\mathbf{W}_V. \tag{21}$$

The intuition behind query and key matrices ($\mathbf{Q}_{(l)}$ and $\mathbf{K}_{(l)}$) is to compute the attention scores $\mathcal{A}_{(l)}$ (Vaswani et al., 2017). Then, $\mathcal{A}_{(l)}$ aggregate information from $\mathbf{V}_{(l)}$,

$$\mathcal{A}_{(l)} = \frac{\mathbf{Q}_{(l)}\mathbf{K}_{(l)}^\top}{\sqrt{d_K}}, \ \mathbf{H}^{(l)} = \text{FFN}(\text{softmax}(\mathcal{A}_{(l)}\mathbf{V}_{(l)})). \tag{22}$$

The definition of $\mathbf{H}^{(l)}$ are same as GCN. softmax is an activation function (Rasamoelina et al., 2020).

## 4 Experiments

### 4.1 Data Construction

The surface code we use can be illustrated as a grid-like structure, as shown in Figure 2. Errors may occur in the data qubits (depicted as circles), which then cause corresponding syndromes in the ancilla qubits (denoted as boxes). Besides, different error types cause the corresponding syndrome through red/blue lines. To evaluate the performance of different machine learning models on quantum error correction, we simulate the errors within the surface code and obtain data for training, validation, and testing. Following Roffe (2019); Chamberland et al. (2022), our data construction process evolves through the following steps: **(1) Error Generation:** $X$ and $Z$ errors in the data qubit are generated based on a probability $p$. In general, the probability $p$ is a small number since the probability that an error occurs is also small. **(2) Syndrome Extraction:** When these errors occur, we employ a stabilizer to identify and extract the corresponding syndrome from the ancilla qubits connected to the data qubits. **(3) Data Structuring:** The detected syndrome serves as the input features for our learning algorithm, with the corresponding errors in the data qubits acting as the output labels. This allows us to learn the error patterns inherent in the surface code. **(4) Handling Degeneracy:** A distinguishing feature of the surface code is its degeneracy, implying that multiple unique error patterns can yield the same syndrome. To manage this, we opt for the set with the

Table 2: Error Correction Rate (%). *p* is the probability of X and Z error in data qubits. OOM is the out-of-memory error. For each task, we **highlight** the best method and underline the statistically significantly better (pass the t-test, i.e., p-value<0.05) methods than CNN, the best existing method (Chamberland et al., 2022).

| Distance | 3 | | | 5 | | | 7 | | |
|---|---|---|---|---|---|---|---|---|---|
| p | 0.005 | 0.01 | 0.05 | 0.005 | 0.01 | 0.05 | 0.005 | 0.01 | 0.05 |
| MWPM | 15.74 | 15.38 | 11.22 | 16.32 | 16.10 | 15.83 | 25.67 | 20.42 | 10.38 |
| CNN | 14.95 | 25.39 | 20.43 | 18.90 | 18.37 | 17.43 | 40.50 | 39.25 | 15.65 |
| U-Net | **96.79** | **93.55** | 71.03 | **97.36** | 93.73 | 70.42 | 97.29 | **95.07** | **69.49** |
| GCN | 89.76 | 87.18 | 67.41 | 95.95 | 93.91 | 66.47 | 96.93 | 92.54 | 68.47 |
| GCNII | 96.69 | 93.22 | **75.14** | 97.29 | **95.85** | 68.67 | 97.51 | 94.33 | 67.16 |
| APPNP | 52.53 | 44.71 | 6.03 | 73.63 | 71.56 | 26.90 | 81.28 | 78.86 | 61.17 |
| Multi-GNN | 81.75 | 78.63 | 56.81 | 88.90 | 79.89 | 61.60 | 95.75 | 89.17 | 59.53 |
| GraphTransformer | 96.70 | 93.41 | 64.94 | 97.14 | 94.79 | **72.47** | **97.53** | 94.14 | OOM |

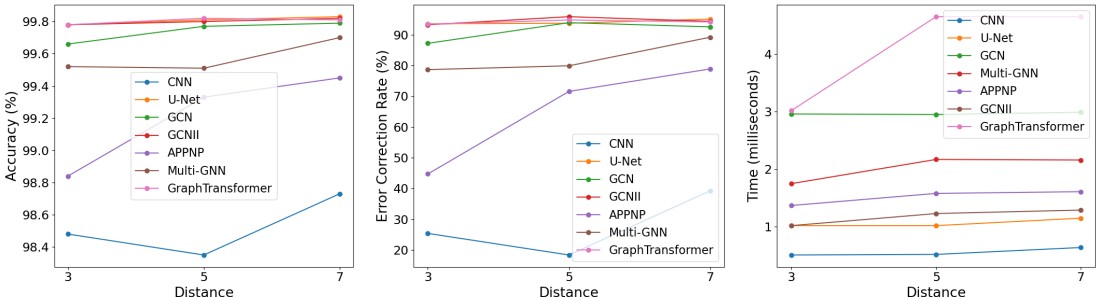

Figure 7: Scalability Analysis. The change of overall accuracy, error correction rate, and inference time as distance grows.

fewest errors for our training input whenever different patterns generate identical syndromes. This selection is underpinned by the assumption that quantum errors are infrequent. However, in the testing phase, we generate the surface code without accounting for this degeneracy. This approach pushes our models to generalize more effectively for the unpredictability inherent in real-world quantum systems.

## 4.2 Experimental Setting

During the training phase, we consider the degenerate nature of the surface code. We vary the distance of surface code from $\{3, 5, 7\}$ (number of qubits is correspondingly 9, 25, 49) to mimic the realistic setup in the current quantum hardware (# of qubit < 50) (Roffe, 2019; Resch & Karpuzcu, 2021; Wang et al., 2022; Gill et al., 2022). We generate 1,000,000 or 10,000,000 surface codes given the distance of a surface code and select the surface codes that meet the degenerate requirement. The details can be found in Appendix A. But in the testing phase, we do not consider this aspect. Therefore, we generate 1,000,000 surface codes without a filter. Our objective is to ensure our approach can model the complexities and unpredictability of real-world quantum errors, thereby improving its generalizability. The statistics of our generated dataset can be found in Appendix A. We classify data qubits into four categories: no error, $X$ error, $Z$ error, and $X/Z$ errors. We leverage two evaluation metrics: (1) overall accuracy and (2) error correction rate. Specifically, overall accuracy refers to the accuracy of "all the data qubits" while the error correction rate refers to the accuracy of "all the data qubits with error". Also, we conduct hypothesis testing to showcase the statistical significance of various methods over CNN (Chamberland et al., 2022), the best existing method. We incorporate minimum weight perfect matching (MWPM) (Criger & Ashraf, 2018) as baseline method, which is the state-of-the-art non-data-driven approach. Implementation details are provided in Section C.1.

Table 3: Inference time (milliseconds). $p$ is the probability of X and Z error in data qubits.

| Distance | 3 | | | 5 | | | 7 | | |
|---|---|---|---|---|---|---|---|---|---|
| p | 0.005 | 0.01 | 0.05 | 0.005 | 0.01 | 0.05 | 0.005 | 0.01 | 0.05 |
| CNN | 0.60 | 0.51 | 0.54 | 0.57 | 0.52 | 0.60 | 0.57 | 0.64 | 0.57 |
| U-Net | 1.32 | 1.02 | 1.18 | 1.06 | 1.02 | 1.00 | 0.99 | 1.15 | 1.14 |
| GCN | 2.49 | 2.96 | 4.46 | 3.03 | 2.95 | 4.37 | 3.15 | 2.99 | 4.51 |
| Multi-GNN | 1.79 | 1.75 | 1.93 | 2.00 | 2.17 | 2.00 | 2.26 | 2.16 | 2.42 |
| APPNP | 1.67 | 1.37 | 1.72 | 1.64 | 1.58 | 1.51 | 1.69 | 1.61 | 1.66 |
| GCNII | 1.16 | 1.02 | 1.13 | 1.34 | 1.23 | 1.21 | 1.26 | 1.29 | 1.33 |
| GraphTransformer | 3.12 | 3.02 | 4.65 | 4.54 | 4.65 | 4.88 | 4.11 | 4.65 | 4.38 |

### 4.3 Results

**Overall Accuracy and Error Correction Rate.** Tables 1 and 2 report the overall accuracy and error correction rate, respectively. We can observe that the overall accuracy of each model is quite close, all exceeding 90%. However, there is a noticeable difference in the error correction rate. This is because most data qubits do not have errors, and predicting labels for error-free data qubits is much simpler than predicting labels for qubits with errors. Compared to conventional CNNs, the error correction rate of U-Net is much higher. This is because U-Net can expand its receptive field through upsampling and downsampling to aggregate more information from distant ancilla qubits for QEC. Different network architectures (CNN, GNN, Transformer) achieved similar error correction rates. This also validates our claim that we need a context-aware model to capture implicit long-range dependencies. We also observed that the model's error correction rate gradually declines as the probability of errors increases. This decline is attributed to the fact that as error probability increases, multiple error configurations corresponding to a single syndrome configuration may emerge during testing. However, our discriminative model can predict at most one of these configurations.

**Inference Time.** We evaluate each model's time cost to infer a surface code and report the mean time of 10,000 experiments. Table 3 shows that the inference time typically falls within milliseconds. Despite the CNNs having more parameters than GNNs, the GNNs' processing time is longer, largely due to the time-intensive aggregation operation. Furthermore, GCN's time cost is more than that of GCNII. This discrepancy can be attributed to our choice of using PyG for GCN's implementation, suggesting that the specific implementation approach can influence inference time cost.

**Scalability Analysis.** We also evaluate the scalability of various methods. We report the change of overall accuracy, error correction rate, and inference time as the distance of the surface code grows in Figure 7. We observe that almost all the methods scale very well. For all the approaches, performance (overall accuracy and error correction rate) would not deteriorate as $d$ (distance) grows. The performance would even increase for some GNN methods (e.g., GCN, APPNP, and Multi-GNN). The key reason is it is easier to mine patterns in larger graphs (Gasteiger et al., 2019). They also exhibit desirable scalability in terms of inference time. That is, growing $d$ would not increase inference time significantly.

## 5 Conclusion

In our study on Quantum Error Correction (QEC) within quantum computing, we highlighted the significance of recognizing the implicit long-range dependencies in the surface code. While traditional QEC methods have their limitations, our machine learning-based approach, informed by these long-range dependencies, offers an improved strategy for error detection. Meanwhile, Our newly designed benchmark evaluates various deep learning algorithms to further support this perspective, emphasizing the potential of distant ancilla qubits in enhancing QEC accuracy. As the frontier of quantum computing expands, our findings provide an essential foundation for future research and optimizations in the domain of QEC.

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

Table 4: Mathematical notations.

| Notations | Explanations |
|---|---|
| $|\psi\rangle$ | quantum state |
| $D_i$ | the $i$-th data qubit |
| $A_i$ | the $i$-th ancilla qubit |
| $\mathcal{G} = (\mathcal{V}, \mathcal{E})$ | an undirected graph representing surface code with distance $d$ |
| $\mathcal{V}$ | set of nodes |
| $\mathcal{E}$ | the set of edges representing the controlled-$X/Z$ operations |
| $d$ | distance of surface code |
| $\mathcal{N}_k$ | neighborhood of node $v_k$ |
| $\Theta$ | Learnable parameter of machine learning model. |
| $\widetilde{\mathbf{Y}}_k$ | the prediction of the $k$-th data qubit |
| $\mathbf{Y}_k$ | groundtruth of the $k$-th data qubit |
| $\mathbf{X}_{\mathcal{N}_k}$ | the syndrome information in the nearby ancilla qubits of the $k$-th data qubit |
| $\mathcal{L}$ | learning objective |
| $\mathbf{I}$ | identity matrix |
| $\mathcal{A}$ | Adjacency matrix of the 2D grid. |
| $\mathbf{H}^{(l)}$ | Hidden state of the graph neural network at the $l$-th layer |
| $\sigma$ | activation function |
| $\widetilde{\mathcal{A}}$ | $\widetilde{\mathcal{A}} = \mathcal{A} + \mathbf{I}$ |
| $\widetilde{\mathbf{D}}$ | diagonal node degree matrix of $\widetilde{\mathcal{A}}$ |
| $\mathbf{W}^{(l)}$ | learnable weight matrix in the $l$-th layer |
| $\mathbf{W}$ | learnable weight matrix |
| $|$ | the concatenation of vectors |
| $\%$ | modulo operator |
| $p$ | the probability of X and Z error in data qubits |

## A  Dataset Details

For the training dataset, we produce either 1,000,000 or 10,000,000 surface codes, selecting the one with the fewest errors for a given syndrome setup. We opt for a larger simulation size of 10,000,000 when the distance is 3 due to the limited number of selected surface codes. For other distances, we generate 1,000,000 surface codes. Table 5 reports the details of the simulation.

Table 5: Data simulation for the training dataset. $p$ is the probability of X and Z error in data qubit.

| Distance | 3 | | | 5 | | | 7 | | |
|---|---|---|---|---|---|---|---|---|---|
| | p=0.005 | p=0.01 | p=0.05 | p=0.005 | p=0.01 | p=0.05 | p=0.005 | p=0.01 | p=0.05 |
| Simulation | 10,000,000 | 10,000,000 | 10,000,000 | 1,000,000 | 1,000,000 | 1,000,000 | 1,000,000 | 1,000,000 | 1,000,000 |
| Training | 1,057 | 1,588 | 4,044 | 10,950 | 41,885 | 619,937 | 50,221 | 179,702 | 969,688 |

The dataset statistics can be found in Table 6.

## B  Additional Empirical Results: Over-Smoothing Analysis for QEC

In graph neural networks such as GCN and GCNII, the input is a graph structure, our goal is to learn a representation for each node in the graph.  Over-smoothing suggests that node representations converge

Table 6: Dataset Statistics. *p* is the probability of X and Z error in data qubit.

| Distance | 3 | | | 5 | | | 7 | | |
|---|---|---|---|---|---|---|---|---|---|
| | p=0.005 | p=0.01 | p=0.05 | p=0.005 | p=0.01 | p=0.05 | p=0.005 | p=0.01 | p=0.05 |
| Training | 1,057 | 1,588 | 4,044 | 10,950 | 41,885 | 619,937 | 50,221 | 179,702 | 969,688 |
| Validation | 1,000,000 | 1,000,000 | 1,000,000 | 1,000,000 | 1,000,000 | 1,000,000 | 1,000,000 | 1,000,000 | 1,000,000 |
| Test | 1,000,000 | 1,000,000 | 1,000,000 | 1,000,000 | 1,000,000 | 1,000,000 | 1,000,000 | 1,000,000 | 1,000,000 |

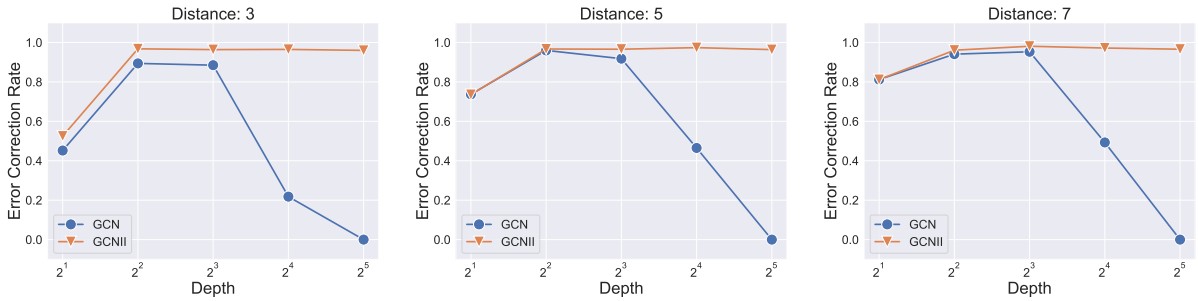

Figure 8: Analysis of the impact of model depth on the performance of GCN and GCNII. We set the error probability as 0.005.

to the same vector and thus become indistinguishable after many-layer graph convolution (Zhao & Akoglu, 2019). In this section, we delve into the influence of depth (number of hidden layers) of GCN and GCNII on QEC's performance. As illustrated in Figure 8, the phenomenon of over-smoothing is evident in QEC as well. As we increase the depth of GCN, its performance peaks and subsequently declines. With a depth of 32 in GCN, the error correction rate drops to zero. However, due to the initial residual connection in GCNII, its performance remains unaffected by over-smoothing. Moreover, we note that as the distance of the surface code grows, both GCN and GCNII require greater depth to attain the best performance. This underscores the need for information from increasingly distant ancilla qubits to predict quantum errors more precisely in expansive surface codes. Consequently, while increasing the number of GNN layers for information aggregation, it is significant to bypass over-smoothing.

## C    Reproducibility

### C.1    Implementation Details

All the models are trained and evaluated using 1 NVIDIA Tesla V100 GPU. We select the hyper-parameters with the validation dataset. The hyper-parameter of our evaluated models can be found in Appendix C.2.

We use Pytorch (Paszke et al., 2019) and PyG (Fey & Lenssen, 2019) to implement our evaluated models. All the experiments are conducted on a single NVIDIA Tesla V100 with 32GB memory size. The softwares that we use for experiments are Python 3.6.8, pytorch 1.9.0, pytorch-scatter 2.0.9, pytorch-sparse 0.6.12, numpy 1.19.2, torchvision 0.10.0, CUDA 10.2.89, CUDNN 7.6.5, einops 0.4.1, and torch-geometric 2.0.3.

### C.2    Hyper-parameter

We tune the hyperparameter with the validation dataset. Table 7, 8, and 9 reports the details of hyper-parameters.

Table 7: The hyper-parameters for CNN, U-Net, and GCN.

| Model | Distance | Probability | Model Type | Hyper-parameters |
|---|---|---|---|---|
| CNN | 3 | 0.005 | CNN | epochs: 1000, lr: 0.01, batch size: 32, optimizer: Adam, kernel size: $3 \times 3$, layers: 1 |
| | 3 | 0.01 | CNN | epochs: 1000, lr: 0.01, batch size: 32, optimizer: Adam, kernel size: $3 \times 3$, layers: 1 |
| | 3 | 0.05 | CNN | epochs: 1000, lr: 0.01, batch size: 64, optimizer: Adam, kernel size: $3 \times 3$, layers: 1 |
| | 5 | 0.005 | CNN | epochs: 1000, lr: 0.01, batch size: 256, optimizer: Adam, kernel size: $3 \times 3$, layers: 1 |
| | 5 | 0.01 | CNN | epochs: 1000, lr: 0.01, batch size: 256, optimizer: Adam, kernel size: $3 \times 3$, layers: 1 |
| | 5 | 0.05 | CNN | epochs: 1000, lr: 0.01, batch size: 1024, optimizer: Adam, kernel size: $3 \times 3$, layers: 1 |
| | 7 | 0.005 | CNN | epochs: 1000, lr: 0.01, batch size: 1024, optimizer: Adam, kernel size: $3 \times 3$, layers: 1 |
| | 7 | 0.01 | CNN | epochs: 1000, lr: 0.01, batch size: 4096, optimizer: Adam, kernel size: $3 \times 3$, layers: 1 |
| | 7 | 0.05 | CNN | epochs: 1000, lr: 0.01, batch size: 4096, optimizer: Adam, kernel size: $3 \times 3$, layers: 1 |
| U-Net | 3 | 0.005 | CNN | epochs: 1000, lr: 0.01, batch size: 32, optimizer: Adam, layers: 2 |
| | 3 | 0.01 | CNN | epochs: 1000, lr: 0.01, batch size: 32, optimizer: Adam, layers: 2 |
| | 3 | 0.05 | CNN | epochs: 1000, lr: 0.01, batch size: 64, optimizer: Adam, layers: 2 |
| | 5 | 0.005 | CNN | epochs: 1000, lr: 0.01, batch size: 256, optimizer: Adam, layers: 2 |
| | 5 | 0.01 | CNN | epochs: 1000, lr: 0.01, batch size: 256, optimizer: Adam, layers: 2 |
| | 5 | 0.05 | CNN | epochs: 1000, lr: 0.01, batch size: 1024, optimizer: Adam, layers: 2 |
| | 7 | 0.005 | CNN | epochs: 1000, lr: 0.01, batch size: 1024, optimizer: Adam, layers: 2 |
| | 7 | 0.01 | CNN | epochs: 1000, lr: 0.01, batch size: 4096, optimizer: Adam, layers: 2 |
| | 7 | 0.05 | CNN | epochs: 1000, lr: 0.01, batch size: 4096, optimizer: Adam, layers: 2 |
| GCN | 3 | 0.005 | GNN | epochs: 1000, lr: 0.01, batch size: 32, optimizer: Adam, layers: 3, hidden: 16 |
| | 3 | 0.01 | GNN | epochs: 1000, lr: 0.01, batch size: 32, optimizer: Adam, layers: 4, hidden: 16 |
| | 3 | 0.05 | GNN | epochs: 1000, lr: 0.01, batch size: 64, optimizer: Adam, layers: 6, hidden: 16 |
| | 5 | 0.005 | GNN | epochs: 1000, lr: 0.01, batch size: 256, optimizer: Adam, layers: 4, hidden: 16 |
| | 5 | 0.01 | GNN | epochs: 1000, lr: 0.01, batch size: 256, optimizer: Adam, layers: 4, hidden: 16 |
| | 5 | 0.05 | GNN | epochs: 1000, lr: 0.01, batch size: 1024, optimizer: Adam, layers: 6, hidden: 16 |
| | 7 | 0.005 | GNN | epochs: 1000, lr: 0.01, batch size: 1024, optimizer: Adam, layers: 4, hidden: 16 |
| | 7 | 0.01 | GNN | epochs: 1000, lr: 0.01, batch size: 4096, optimizer: Adam, layers: 4, hidden: 16 |
| | 7 | 0.05 | GNN | epochs: 1000, lr: 0.01, batch size: 4096, optimizer: Adam, layers: 6, hidden: 16 |

Table 8: The hyper-parameters for MultiGNN, APPNP, and GCNII.

| Model | Distance | Probability | Model Type | Hyper-parameters |
|---|---|---|---|---|
| MultiGNN | 3 | 0.005 | GNN | epochs: 1000, lr: 0.01, batch size: 32, optimizer: Adam, layers: 5, hidden: 16 |
| | 3 | 0.01 | GNN | epochs: 1000, lr: 0.01, batch size: 32, optimizer: Adam, layers: 5, hidden: 16 |
| | 3 | 0.05 | GNN | epochs: 1000, lr: 0.01, batch size: 64, optimizer: Adam, layers: 7, hidden: 16 |
| | 5 | 0.005 | GNN | epochs: 1000, lr: 0.01, batch size: 256, optimizer: Adam, layers: 5, hidden: 16 |
| | 5 | 0.01 | GNN | epochs: 1000, lr: 0.01, batch size: 256, optimizer: Adam, layers: 5, hidden: 16 |
| | 5 | 0.05 | GNN | epochs: 1000, lr: 0.01, batch size: 1024, optimizer: Adam, layers: 7, hidden: 16 |
| | 7 | 0.005 | GNN | epochs: 1000, lr: 0.01, batch size: 1024, optimizer: Adam, layers: 7, hidden: 16 |
| | 7 | 0.01 | GNN | epochs: 1000, lr: 0.01, batch size: 4096, optimizer: Adam, layers: 7, hidden: 16 |
| | 7 | 0.05 | GNN | epochs: 1000, lr: 0.01, batch size: 4096, optimizer: Adam, layers: 7, hidden: 16 |
| APPNP | 3 | 0.005 | GNN | epochs: 1000, lr: 0.01, batch size: 32, optimizer: Adam, $K$: 5, $\alpha$: 0.5, hidden: 16 |
| | 3 | 0.01 | GNN | epochs: 1000, lr: 0.01, batch size: 32, optimizer: Adam, $K$: 5, $\alpha$: 0.5, hidden: 16 |
| | 3 | 0.05 | GNN | epochs: 1000, lr: 0.01, batch size: 64, optimizer: Adam, $K$: 5, $\alpha$: 0.5, hidden: 16 |
| | 5 | 0.005 | GNN | epochs: 1000, lr: 0.01, batch size: 256, optimizer: Adam, $K$: 5, $\alpha$: 0.5, hidden: 16 |
| | 5 | 0.01 | GNN | epochs: 1000, lr: 0.01, batch size: 256, optimizer: Adam, $K$: 5, $\alpha$: 0.5, hidden: 16 |
| | 5 | 0.05 | GNN | epochs: 1000, lr: 0.01, batch size: 1024, optimizer: Adam, $K$: 5, $\alpha$: 0.5, hidden: 16 |
| | 7 | 0.005 | GNN | epochs: 1000, lr: 0.01, batch size: 1024, optimizer: Adam, $K$: 5, $\alpha$: 0.5, hidden: 16 |
| | 7 | 0.01 | GNN | epochs: 1000, lr: 0.01, batch size: 4096, optimizer: Adam, $K$: 5, $\alpha$: 0.5, hidden: 16 |
| | 7 | 0.05 | GNN | epochs: 1000, lr: 0.01, batch size: 4096, optimizer: Adam, $K$: 5, $\alpha$: 0.5, hidden: 16 |
| GCNII | 3 | 0.005 | GNN | epochs: 1000, lr: 0.01, batch size: 32, optimizer: Adam, $\alpha$: 0.1, $\beta$: 0.5, hidden: 16, layers: 4 |
| | 3 | 0.01 | GNN | epochs: 1000, lr: 0.01, batch size: 32, optimizer: Adam, $\alpha$: 0.1, $\beta$: 0.5, hidden: 16, layers: 4 |
| | 3 | 0.05 | GNN | epochs: 1000, lr: 0.01, batch size: 64, optimizer: Adam, $K$: 5, $\alpha$: 0.5, hidden: 16 |
| | 5 | 0.005 | GNN | epochs: 1000, lr: 0.01, batch size: 256, optimizer: Adam, $\alpha$: 0.1, $\beta$: 0.5, hidden: 16, layers: 5 |
| | 5 | 0.01 | GNN | epochs: 1000, lr: 0.01, batch size: 256, optimizer: Adam, $\alpha$: 0.1, $\beta$: 0.5, hidden: 16, layers: 5 |
| | 5 | 0.05 | GNN | epochs: 1000, lr: 0.01, batch size: 1024, optimizer: Adam, $\alpha$: 0.1, $\beta$: 0.5, hidden: 16, layers: 5 |
| | 7 | 0.005 | GNN | epochs: 1000, lr: 0.01, batch size: 1024, optimizer: Adam, $\alpha$: 0.1, $\beta$: 0.5, hidden: 16, layers: 5 |
| | 7 | 0.01 | GNN | epochs: 1000, lr: 0.01, batch size: 4096, optimizer: Adam, $\alpha$: 0.1, $\beta$: 0.5, hidden: 16, layers: 5 |
| | 7 | 0.05 | GNN | epochs: 1000, lr: 0.01, batch size: 4096, optimizer: Adam, $\alpha$: 0.1, $\beta$: 0.5, hidden: 16, layers: 5 |

Table 9: The hyper-parameters for Graph Transformer.

| Model | Distance | Probability | Model Type | Hyper-parameters |
|---|---|---|---|---|
| | 3 | 0.005 | Transformer | epochs: 1000, lr: 0.01, batch size: 32, optimizer: Adam, $\alpha$: 0.1, $\beta$: 0.5, hidden: 16, layers: 3 |
| | 3 | 0.01 | Transformer | epochs: 1000, lr: 0.01, batch size: 32, optimizer: Adam, heads: 4, hidden dropout: 0.1, hidden: 16, layers: 3 |
| | 3 | 0.05 | Transformer | epochs: 1000, lr: 0.01, batch size: 64, optimizer: Adam, heads: 4, hidden dropout: 0.1, hidden: 16, layers: 5 |
| | 5 | 0.005 | Transformer | epochs: 1000, lr: 0.01, batch size: 256, optimizer: Adam, heads: 4, hidden dropout: 0.1, hidden: 16, layers: 5 |
| Graph Transformer | 5 | 0.01 | Transformer | epochs: 1000, lr: 0.01, batch size: 256, optimizer: Adam, heads: 4, hidden dropout: 0.1, hidden: 16, layers: 5 |
| | 5 | 0.05 | Transformer | epochs: 1000, lr: 0.01, batch size: 1024, optimizer: Adam, heads: 4, hidden dropout: 0.1, hidden: 16, layers: 5 |
| | 7 | 0.005 | Transformer | epochs: 1000, lr: 0.01, batch size: 1024, optimizer: Adam, heads: 4, hidden dropout: 0.1, hidden: 16, layers: 5 |
| | 7 | 0.01 | Transformer | epochs: 1000, lr: 0.01, batch size: 4096, optimizer: Adam, heads: 4, hidden dropout: 0.1, hidden: 16, layers: 5 |
| | 7 | 0.05 | Transformer | epochs: 1000, lr: 0.01, batch size: 4096, optimizer: Adam, heads: 4, hidden dropout: 0.1, hidden: 16, layers: 5 |

