# OpenReview forum: "Benchmarking Machine Learning Models for Quantum Error Correction"
_TMLR — Rejected by TMLR_

### Review · Reviewer_Xwtb · 2023-12-11

**Summary Of Contributions:**

This paper benchmarks different machine learning models for addressing the problem of quantum error correction (QEC) on quantum computing systems.
The main contribution of the paper are summarized in page 2. Specifically, the novelties of the paper include
- i) the first benchmark of well established deep learning frameworks for QEC task
- ii) highlighing the importance of long range correlations in ancilla qubits in the surface code representation of a system of qubits
- iii) extensive experiments for establishing the advantages and disadvantages of different deep learning (DL) approaches for the task at hand.

While I beleieve there’s a strong overlap between contribution i) and iii) I indeed recognize the novelty of this work in the first two contributions highlighted by the authors.

Despite not being an expert in the field of DL for QEC, I found this paper to be one of the first work making the effort to perform a deeper study in this context benchmarking several DL architecthures for the QEC task.

**Audience:**

Yes

**Broader Impact Concerns:**

I have no broader impact concerns. However, in terms of broader impact, it’d be interest to have an outlook section where the authors discuss the applicabiliy of the approaches they analyze in problems/systems of actually relevant sizes.

**Final Recommendation**

As remarked several time,  the paper is scientifically valid and may have a relevant impact in the field as it tackles a highly relevant problem in QC using ML.
However, because of the **Weaknesses** listed above as well as the several concerns raised in the **Requested changes** section, I have to refrain from recommending publication of this paper until a major revision is done by the authors.

**Claims And Evidence:**

Yes

**Requested Changes:**

In this section, the necessary changes for the paper will be discussed:

- As noted in the **Weaknesses** section above, the structure requires some adjustments. Specifically, my suggestion would be to reduce the Introduction section to a general discussion of the broader impact of QEC and how ML can improve upon standard approaches. This should be followed by bullet points outlining the contributions of the paper. After this short introduction, basics (e.g., section 2) should follow. The methods discussed in the current introduction (which somehow assumes some concepts such as syndrome and surface code to be known to the reader) should then appear as a later subsection, e.g., 2.6 Related Work or similar. This would make it easier for readers familiar with QC but less acquainted with the techniques for QEC discussed in this paper to follow. One notable example is that the introduction is not fully self-contained. The notion of the syndrome of an Ancilla qubit and the concept of the surface code are not introduced appropriately (unless I overlooked this in the paper).

- Some paragraphs are hard to parse and difficult to follow, requiring a major revision. Specifically, I was particularly confused by Sec. 2.5, especially the paragraphs between Eqs. 13 and 14. I found the notation with % (the modulo operation(?)) not introduced yet not clear. I completely missed the meaning of Eq. 13 and the following description. I would, therefore, appreciate it if the authors could carefully revise this section, expanding the discussion and carefully introducing all the notation without omitting any detail, even if that may seem obvious at first glance.

- I found some sentences/paragraphs to be unclear or imprecise. I hereby make some suggestions for what I found to be more critical:
  - “The most significant advantage of quantum computers over traditional computers is their superior computing power” -> This claim sounds a little bit too vague. It would be great if the authors could expand on this, being more specific and/or adding some relevant references.
  - “To get the computation results, a qubit readout measures the probability of a qubit state |ψ⟩ collapsing into one of its basis states, either |0⟩ (yielding output y = +1) or |1⟩ (yielding output y = −1). The probabilities of yielding |0⟩ and |1⟩ are |α|2 and |β|2, respectively.” -> This sentence comes after the general case for systems of N qubits is discussed. However, this deals again with a single-qubit system. This might be counterintuitive and possibly confusing. I’d recommend putting this earlier or adjusting it to the N qubit systems.
  - “Unlike error-type, which is the bit-flip in traditional bits, the error types in qubits are infinite due to their stochastic nature” -> I found this sentence unclear too.

- Concerning the structure of the paper, Figures might be better placed. For example, something I found particularly unfortunate is that the paper often refers to Figure 2, even on page 6 though the figure appears on page 4. This made it a bit difficult to navigate the paper.

- As far as the ML part is concerned (Sec. 3-4), I found some details to be missing from the main discussion. Specifically, while it is clear that this is a supervised learning approach, it is not clear what the considered objective is and how the models are trained. While this information may be hidden in the text, I believe the paper would benefit from making this explicit. This problem also somehow entangles with the upcoming point (see below).

- I found paragraph 3.2 to be a bit disconnected from the rest of the paper. While the probabilistic view naturally fits the discussion, it is not apparent how to relate this to the upcoming section. Specifically, I have a problem connecting (on a more formal, mathematical standpoint) eq. 16 to the following equations describing the updating rule for each proposed ML model. Specifically, what are the parameters $\Theta$ from eq. 15 and 16? Do they represent the parameters of the models’ networks (e.g., the weight matrices)?

- Regarding the ML models introduced in Sec. 3, I think many require a more careful discussion, and some notations are not always consistent. I suggest a major revision of this section. See also the other points highlighting the problem with respect to figure 5 and related points in Sec. **Minor Issues** below.

- Captions are sometimes not self-contained. For instance, captions of Tables 1 and 2 do not mention what p is. While it can be apparent, I’d recommend always ensuring that every caption allows full interpretation of every detail of the plot to enhance the readability of the manuscript. A major problem is represented by the caption of Table 3, which gives no information at all.

- Figures need some work. For example, if someone reads the paper in grayscale, it is very hard to distinguish between red and blue lines. This may also represent an issue for colorblind people. I, therefore, recommend representing the edges for the controlled-X and controlled-Z with dashed and solid lines, for instance, instead of blue and red. Moreover, I have some difficulty appreciating the reason for other figures, e.g., Figure 5. This latter just sketches input/output pairs while the GCN structure is completely omitted. This graphic does not add any information on top of what’s already shown in Figure 4 and others. Instead, it would be more useful to have a sketched representation or a cartoon of a GCN and GCNII. I think it would also be important, within the context of CNN, to show a sketch of those for people who are not familiar with ML/DL concepts.

- Some references seem to be missing or at least some are a bit outdated. I’d recommend the authors go through the paper and add references where appropriate. For example, all the refs in the introduction motivating the potential impact of QC are too few and a little outdated. Moreover, when the DL models are introduced later in the paper, many references to recent works and applications (especially outside the context of QEC) are missing.

- The experiments represent one of the major flaws in this paper. I believe this section may need some work, and I think the paper would greatly benefit from the following:
  - Benchmarking using standard methods among the DL models presented in the paper. Not sure what the state-of-the-art (SOTA) is there, but I’d choose at least one. This would allow appreciation of the benefits of ML methods in contrast to non data-driven approaches.
  - I did not fully get the difference between Overall accuracy and the Error correction rate. Broadening the discussion there would be crucial for letting the reader understand the results better.
  - I do not fully get the point for Table 3. If the authors thought of this as being a metric of success, they may also need to broaden the discussion, including an analysis of the training time, not only for inference. Eventually, for a fair comparison, what matters is the total time as the sum of the two (training+inference).
  - How is the result for the best method obtained? Is it the mean of an ensemble of trained models with different seeds? Or is it just a single model? It would be important to broaden the analysis and get some statistical errors over those values. Many of those results are very close, which suggests the models might be statistically equivalent when considering the corresponding uncertainties.

**Conceptual Questions**

Here, I'd like to pose some conceptual questions to engage the authors and prompt further analysis, either in a revision of this manuscript or in future works:

- Has the author conducted a comprehensive scalability analysis? How do the models scale? Does performance deteriorate as d (distance) increases? What is the threshold beyond which performance substantially deteriorates? Can any scaling law be derived from this? How does this scaling law compare to standard methods (perhaps MWPM)?

- Have the authors considered the possibility of using parameter sharing to reuse models trained on smaller distances (e.g., smaller surface code settings) for capturing correlations at larger distances? Some benefits can be expected, especially if the goal is to scale to very large systems.

- What would be the required distance (surface code extent) to apply the present algorithms to realistic problems on current hardware? A broader discussion on this would be helpful to understand the limitations of the analyzed ML approaches.

- Do these systems exhibit any symmetries that one can naturally leverage to enhance the training of better models, possibly on fewer data?

- *Naive question to the authors*: Is this the first work highlighting the importance of longer-range correlations between ancilla qubits, or was this already established in the field and confirmed by achieving better results using more context-aware models such as UNets?

**Minor Issues**
- “will release the code when the paper is published”: I’d remove this sentence from the abstract and instead put a link to a repository/code in the later version of the paper.
- Above equation 1: I recommend to substitute “as follows” with “as a quantum supersposition of states”
- Above equation 2: I recommend to substitute “as follows” with “as a tensor product”
- Above equation 3: I recommend to remove “as follows”
- Caption of Figure 1: one of the \psi_L\lrangle is missing a \vert (end of third sentence of the caption).
- Before equations 18 and 19 the paper says “[…] The l-th layer can be formulated/written as follows”; however, eq 19 shows layer l+1 while eq 19 does not show layer l at all. I recommend the authors to carefully proofread the math of Section 3 as it seems to contain several flaws.
- The referencing of the appendix seems to be broken as it appears as ??.
-  What is the meaning for the subscript ‘fc’ in equation in eq. 20?
- Above equation 21 the authors say “we employ query, key and value matrices […]”. What are those, especially in this context? What is their meaning? I think we should not assume every reader is familiar with Transformers architechtures (and the other models) and therefore Sec. 3 requires a more careful revision when introducing each model.
- “Over-smoothing suggests that node representations converge to the same vector and thus become indistin- guishable after many-layer graph convolution” -> I may have missed this in the paper, but what do the author mean with ‘node representations’?

**Strengths And Weaknesses:**

**Strengths**

- The paper is generally well-written and pleasant to read.
- Although some sections are harder to parse, and there are occasional imprecise sentences, the overall scientific content is sound and potentially interesting to a broad audience.
- The paper addresses a pivotal issue for NISQ-era devices, making it highly relevant for researchers in both machine learning (ML) and quantum computing (QC) domains.
- Highlighting the importance of long-range correlations of syndromes in ancilla qubits for identifying data qubits with errors it's an important contribution of this work.

**Weaknesses**

- The paper suffers from technical and structural flaws requiring a major revision before recommending publication.
- Structural issues include the need to adjust the Introduction section, anticipate broader QC and QEC discussions, and postpone the Related Work section.
- Technical flaws involve vague or imprecise sentences and some redundancy in the writing style.
- Figures and captions need adjustments for clarity and self-containment.
- The experiments section lacks clarity and precision, requiring detailed analysis and improvement.
- Minor typos and other issues, including missing or outdated references, need to be adjusted.

---

> ### Comment · Reviewer_Xwtb · 2024-01-02
> **Revised Manuscript**
>
> I’d like to thank the authors for their efforts in revising the paper and implementing many of the changes requested by the referees. At this stage, it already looks much improved. Nevertheless, I still have a few remarks and questions for the authors:
>
> - I am struggling to fully grasp the details of Sec. 2.5. Specifically, I do not fully understand Eq. (13). As per my understanding, $k$ represents the position on the graph and does not refer to the measurement whatsoever. The first two rows represent the case where the node is an ancilla qubit (odd sites, k%2=1) and a data qubit with no syndrome (even sites, k%2=0) respectively. However, I am not entirely sure how to interpret rows 3 and 4. Again, to my understanding, in both rows, the k-th node ($v_k$) has a syndrome, but I am not entirely sure about the first condition. I would appreciate it if the authors could further elaborate on this notation and describe it more intuitively. What is the meaning of the lower corner square brackets?
>
> - I agree with point 4 in the **requested changes** from reviewer seGs. Specifically, I think it is still not clear from the main text that $Y_k$ represents valid probabilities.
>
> - In the caption of Table 1, the authors write, *"We underline the statistically significantly better (pass the t-test, i.e., p-value<0.05) methods than CNN, the best existing method (Chamberland et al., 2022).”* I found this a bit hard to parse. As an alternative, I suggest highlighting the CNN by clearly stating that this is the previous state of the art. In a separate sentence, mention that the methods passing the t-test and being better than SOTA (CNN) are underlined. Finally, represent the best results in bold.
>
> - In Table 1, there are no errors in the overall accuracies. I wonder if the authors have tried to train different models (with different seeds) and checked whether the performance is consistent. This can be crucial since, for d=5 and p=0.005, the overall accuracy of GCNII, GCN, and U-Net are very close to each other. Considering an ensemble of methods might show some fluctuations, thus making the outcomes for the different architectures statistically compatible. Similar considerations for this and the previous bullet point apply to Table 2.
>
> - To enhance the readability of Fig. 7, I suggest creating one legend and placing it once outside the plots to avoid overlap, which might hinder the reading of the plot. It would also be helpful to have error bars in Fig. 7 (similar comment that applies to the Tables).
>
> - Table 3 does not show any statistical evaluations for the uncertainty. Here, it would be sufficient to report the standard deviation along with the mean for the 10,000 experiments. Also, I do not understand where the authors obtained the inference time for GraphTransformer for d=7 and p=0.05 since this setting led to OOM issues in previous tables.

---

### Review · Reviewer_UJcB · 2023-12-24

**Summary Of Contributions:**

In this paper the authors benchmark the performance of various ANNs (Artificial neural Networks) in the task of detecting errors on qubits. The paper is well written and provides sufficient background to both quantum computing as well as in the current error correction within the area of machine learning. The author provide a set of features allowing to formalize the 2D planar map of a set of qubits connected to ancillae qubits and perform comparative experiments on both the detection and the correction.

**Audience:**

No

**Claims And Evidence:**

Yes

**Requested Changes:**

- The evaluation must be much more detailed as written in the Strengths and Weaknesses
- The \em syndrome \em should be more explained in particular what are the values it takes for different error, and how it shows degeneracy (before constructing the features input)
- More detailed analysis of the resulting accuracy using tools such as confusion matrices and per class accuracy detailed information

**Strengths And Weaknesses:**

Strengths:

- The authors provide a formulation of the extraction of information about faulty qubits and present it as a structured input to a machine learning interface.

Weaknesses:

- I feel certain details are not explained in detailed fashion required for full understanding. For instance what is the purpose of equation 12? As far as I can tell the information from eq.12 is not used anywhere else such as for instance in the feature generation (eq. 13)
- The evaluation. The paper is entitled "Benchmarking...." however all it present is the comparison of a selected algorithms on a set of randomly and very roughly statistics based controlled data. As benchmarking goes there is not much benchmarking as there is no analysis of the data and of the performance of the models such as what is the distribution of errors , what is their mixture and how individual models are solving the various combinations. Therefore the contribution of the paper i minimal from the point of view of generalizing research because it falls short of explaining what actually the results mean.
- The authors never explain what they mean by syndrome. While it comes out implicitly after reading the paper such keywords should be defined more precisely and not assume that the reader knows the meaning in the context
- The evaluation. the test dataset contains by default a very small amount of errors given the low probabilities of injecting the noise. However there is no discussion about how the authors ensure that the measure of the correctness are normalized so that for instance the 90% of accuracy does not mean that the network is not simply labeling all the outputs by the majority class label.
- Some English check, For instance Section 2.1 is entitled \em Quantum Basic \em and I think it should be \em Quantum Basics\em.

---

### Review · Reviewer_seGs · 2023-12-27

**Summary Of Contributions:**

The submission concerns itself with developing a benchmark for the problem of quantum error correction (QEC) via machine learning, with a specific focus on also evaluating the importance of information about long-range ancilla qubits for accurate error prediction.
The authors claim that ML models for QEC so far have only focused on close-range dependencies between the ancilla and data qubits, so they design a benchmark dataset that can be used to assess the ability of a ML model of exploiting the information in these long-range dependencies. Finally, they evaluate a range of convolution and graph-based neural network models that are capable of capturing the long-range information to various degrees, and show that this additional context about long-rang ancilla qubits does indeed lead to significant improvements in quantum error classification accuracy.

**Audience:**

Yes

**Broader Impact Concerns:**

There are no significant broader impact concerns.

**Claims And Evidence:**

Yes

**Requested Changes:**

1. The authors should include an explanation of the terms syndrome and ancilla qubit in section 2.3, since these terms are not clear to readers who come outside the field of quantum computing.

2. The text in section 2.4 refers to Figure 2, but only to the left subfigure, and this is not made clear anywhere in the text or the figure caption. I would suggest for the authors to annotate the two subfigures as 2a and 2b for better clarity, and refer to them as such in the text and captions. Additionally, the right subfigure in figure 2 is never referred to in the main text, so it is unclear if it is even necessary to include it at all. Additionally, the authors should clearly explain that the red squares in the figure indicate syndromes in the ancilla qubits.

3. In section 2.5, either the explanation about the data qubit features is worded confusingly, or there is a typo in the definitions. More specifically, the authors seem to define data qubits as those nodes where k%2=1, however, on the first line of page 6 the following text "To represent the labels of data qubits (specifically when k%2 = 0)" seems to contradict this.

4. The whole section 3.2 seems redundant from a machine learning perspective. It is trivial to rewrite the conditional probably to extend to other nodes outside of the neighborhood, and the authors do not really do anything with this equation to provide new insights into the problem. Additionally, the definition of the cross entropy loss is written as if the outputs $Y_k$ are probabilities themselves, but from the previous definition of the conditional probability this is not obvious. The authors should specify that $Y_k$ and the corresponding predicted values are guaranteed to be valid probabilities, e.g. since the network uses a softmax for the outputs.

5. I am not familiar with how long it would take to generate 1 or 2 datasets with distances > 7, and how expensive it would be to evaluate the models in this setting. However, it would be important to show that these models are future proof by showing that they scale favorably even for quantum systems of sizes beyond today's quantum hardware.

6. One final point that should be addressed is the possible inconsistency of the inference times presented in Table 3. The authors should make sure that the different implementations used for the different GNN models do not account for a large part of the variation between the inference times for the different models.

**Strengths And Weaknesses:**

Strengths:
- The insight that current models do not sufficiently account for the long-range interactions withing QEC, and the development of the benchmark and evaluation of models to demonstrate this dependence is a valuable contribution.
- The paper is well structured, with the earlier sections explaining the necessary concept for the later sections.
- The benchmark is done with a good range of graph neural network architectures, giving a good base for the most promising directions of future research for better models.

Weaknesses:
- While the paper is well structured, the explanations of the different concepts can be lacking and make understanding the paper more difficult. For example, there are no definitions of the concepts of a syndrome and ancilla qubit, although there is an entire section dedicated to explaining the different aspects of quantum error correction. I have included some more such errors or unclear statements in the requested changes section.
- The benchmarking dataset seem a bit inadequate, especially in terms of scalability and future proofing. The authors claim that such methods should be scalable, but do not test their models on systems that go beyond the scale of current quantum computers in the number of qubits, which would be necessary to demonstrate that these methods will scale well even for system sizes that go beyond the limit of today's quantum computing technology.

---

### Decision · Action_Editor_YTfR · 2024-05-16

**Recommendation:** Reject

**Comment:**

See "Claims And Evidence".

**Audience:**

Yes.

**Claims And Evidence:**

The reviewers raised several questions that are not addressed by the authors. So we think the paper need another round of major revision to improve the experimental evaluation and the clarity of the paper. In prarticular:

1. The scalability of the benchmarking dataset is not well supported, and some details about the experiments are lacking (see Reviewer UJcB, weakness 4). Additionally, as this is a benchmarking paper, the authors should provide justification for why the datasets used are representative.
2. Some clarification questions (e.g., about equation 12) are not addressed.
3. Several questions about Table 3 (from Reviewer seGs and Reviewer Xwtb) need to be addressed. Further, there are several second round questions from Reviewer Xwtb.

Due to the lack of engagement from the authors and the remaining unanswered questions, we believe the paper requires major revisions to address all the reviewers' concerns.

**Resubmission Of Major Revision:**

The authors may consider submitting a major revision at a later time.